**Quantifying Restoration Success of Wood Introductions to Increase Coho Salmon Winter Habitat**

Russell T. Bair[1], Catalina Segura[1,*] and Christopher M. Lorion[2]

[1]Forest Engineering, Resources, and Management, Oregon State University, 201 Peavy Hall Corvallis Oregon 97331 USA

[2]Oregon Department of Fish & Wildlife, 4034 Fairview Industrial Drive SE, Salem, OR 97302 USA

*Correspondence to*: Catalina Segura (segurac@oregonstate.edu)

**Abstract.** Large wood (LW) addition is often part of fish habitat restoration projects. However, there is limited information about the spatial-temporal variability in hydraulic changes after LW additions. We investigated reach scale hydraulic changes triggered after the addition of LW that are relevant to juvenile Coho Salmon survival. We used Nays2DH, an unsteady two-dimensional flow model to quantify patterns and magnitudes of changes of stream velocity and shear stress in three alluvial gravel reaches. The study sites are located in low gradient reaches draining 5 to 16 km$^2$ in the Oregon Coast Range. Survivable habitat was characterized in terms of critical swim speed for juvenile Coho and bed stability considering the critical shear stress required to mobilize the median bed particle size. Model predictions indicated that survivable habitat during bankfull conditions, measured as the area with velocity below the critical swim speed for juvenile Coho, increased by 95–113% after the LW restoration. Bed stability also increased between 86–128% considering the shear stress required to mobilize the median bed particle size. Model predictions indicated more habitat created in the larger site, however considering that wood would move more frequently in this site there appears to be a trade-off between the timing and the resilience of restoration benefits. Overall, this study quantifies how the addition of LW potentially changes stream hydraulics to provide a net benefit to juvenile salmonid habitat. Our findings are applicable to stream restoration efforts throughout the Pacific Northwest.

## 1 Introduction

Large wood (LW) is a fundamental component of many temperate streams given its influence on flow resistance, stream morphology, sediment transport, nutrient cycling, and stream habitat (e.g., Triska and Cromack, 1980; Harmon et al., 1986; Montgomery et al., 1995; Kail, 2003). LW structures increase heterogeneity in the flow field by promoting local scour and sediment retention, by reducing average flow velocity, by influencing bed texture (Buffington and Montgomery, 1999a), and by promoting increased interaction of the flow with the floodplain (Beschta, 1979; Harmon et al., 1986; Lisle, 1986; Bisson et al., 1987; Wipfli et al., 2007; Seo et al., 2008). LW jams are often associated with forced pool-riffle morphologies in reaches that would otherwise exhibit plane-bed characteristics (Montgomery and Buffington, 1997). Thus, channels with abundant LW have relatively higher complexity (e.g., high frequency of pools, channel bars, and riffles), offering a wide range of habitat for aquatic species including invertebrates and fish (Fausch and Northcote, 1992; Gerhard and Reich, 2000; Roni and Quinn, 2001;

Dolloff and Warren, 2003; Jahnig and Lorenz, 2008; Benke and Wallace, 2010; Pess et al., 2012). Historically, abundant LW in Pacific Northwest streams provided habitat for a variety of fish species (Bisson et al., 1988; Connolly and Hall, 1999) including anadromous fish such as Coho Salmon (*Oncorhynchus kisutch*) and steelhead (*Oncorhynchus mykiss*) (Nickelson et al., 1992a; Quinn and Peterson, 1996; Beechie and Sibley, 1997; Johnson et al., 2005; Gallagher et al., 2014; Jones et al., 2014). Prior to the recognition of the role of LW pieces in habitat, forest management operations allowed harvesting to the edge of streams and the removal of in-channel LW. This removal resulted in the reduction of stream complexity (Bisson et al., 1987; Sedell et al., 1988; Stednick, 2008), which has reduced habitat and contributed to fish population declines (Dolloff, 1986; House and Boehne, 1986; Fausch and Northcote, 1992; Smith et al., 1993a; Smith et al., 1993b; Brown et al., 1994).

For Coho Salmon, which generally spend at least one year rearing in freshwater prior to out-migration to the ocean, overwinter survival has been identified as a critical factor influencing population abundance and productivity (Tschaplinski and Hartman, 1983; Nickelson et al., 1992a; Nickelson et al., 1992b; Quinn and Peterson, 1996; Huusko et al., 2007; Gallagher et al., 2012; Suring et al., 2012). Coho Salmon overwinter survival is strongly linked to the availability of complex, low velocity habitats that have been reduced in many areas due to land use and development (Tschaplinski and Hartman, 1983; McMahon and Hartman, 1989; Quinn and Peterson, 1996; Johnson et al., 2005). Thus, restoration of winter refuge habitat for Coho Salmon can be crucial for population viability and species recovery (Nickelson and Lawson, 1998; NMFS, 2016).

The rationale behind LW restoration projects is that the introduced pieces would create larger and deeper pools, stabilize stream substrate, and facilitate the interaction of the flow with the floodplain. This ultimately provides low velocity refuge where juvenile salmonids can shelter both in the stream channel and in adjacent, newly connected floodplains (Bustard and Narver, 1975b; McMahon and Hartman, 1989; Bradford et al., 1995; Cunjak, 1996). However, there is still controversy about the effectiveness of adding LW as a restoration strategy (Roni et al., 2008; Whiteway et al., 2010; Roni et al., 2014). Studies have reported improvements in fish abundance after LW introductions in relatively short reaches (75–500 m) (e.g., House and Boehne, 1986; Cederholm et al., 1997; Roni and Quinn, 2001) while others working over larger scales (500–1000 m) have observed positive changes to stream morphology relevant to fish habitat (Anlauf et al., 2011; Jones et al., 2014). The survey approaches used in these studies provide a static perspective on stream habitat and often occur under low flow conditions. We currently lack understanding of how LW structures affect flow hydraulics and fish habitat at the reach scale under a range of flows, which is relevant to those looking to address both geomorphic change and natural habitat limitations.

Previous efforts have used computational fluid dynamics models to simulate field conditions around obstacles such as wood and boulders in theoretical domains (Allen and Smith 2012) and experiment flumes (Xu and Liu, 2016; Lai et al., 2017; Xu and Liu, 2017) in some cases using flow deflectors to mimic the effects of wood in channels (Biron et al., 2009). These studies have provided detailed descriptions of the turbulent flow around these structures, highlighting the effects of simplifying the geometry of the obstacles in the prediction of flow velocity (Allen and Smith 2012; Xu and Liu, 2017), and the effects of the assumed obstacle shape and orientation on the velocity field and sediment transport (Biron et al., 2009; Biron et al., 2012). However, these models are computational intensive and not yet feasible at the reach scale.

Two-dimensional (2D) computational hydraulic modelling offers a relatively time and cost-effective strategy to analyse the flow field of a stream reach without the need for high-resolution field measurements at every discharge level of interest. These 2D models have been used to quantify fish habitat based on flow velocity and depth indicators in streams in a variety of conditions (e.g., Nagaya et al., 2008; Branco et al., 2013; Cienciala and Hassan, 2013; Hatten et al., 2013; Laliberte et al., 2014; Fukuda et al., 2015; Carnie et al., 2016) including the effects of boulders in straight urban sections (Lee et al., 2010) and the effects of large wood using non-calibrated models (Hafs et al., 2014; Wall et al., 2016). The 2D estimates of velocity and channel bed stability, at scales of ecological significance —individual boulders and LW pieces (Crowder and Diplas, 2000) —can be used to estimate habitat improvements after the addition of LW. Flow velocity can limit the ability of fish to maintain position and result in excessive energetic costs (Huusko et al., 2007), while unstable sediment limits the ability of juveniles to find shelter within substrate rocks during high flows.

Despite the mentioned applications of 2D hydraulic modelling, there are limited examples of calibrated efforts that have evaluated winter habitat for salmonids at the reach scale. Our objective was to use a calibrated 2D model to quantify the change in survivable habitat area for juvenile Coho Salmon after the addition of LW by examining changes in water velocity and substrate stability during a bankfull event in three gravel-bed reaches. In doing so, we developed field calibrated before and after models to describe flow hydraulics in three individual sites. To our knowledge, this was the first time a calibrated model has been used to estimate the effects of LW in natural conditions.

## 2 Methods

### 2.1. Study Area

This study was conducted in three alluvial stream reaches in Mill Creek, a tributary of the Siletz River in the Oregon Coast Range (Fig. 1). The watershed is dominated by intensively managed Douglas fir (*Pseudotsuga menziesii*) forest, and riparian areas are mostly vegetated with the deciduous species vine maple (*Acer circinatum*), bigleaf maple (*Acer macrophyllum*), and red alder (*Alnus rubra*). Watershed elevations range from 60 m to 730 m (Fig. 1) and the basin is primarily underlain by the Tyee formation composed of sandstone and siltstone. The climate is marine temperate, influenced by moisture from the Pacific Ocean, and annual precipitation of 2,300 mm in the nearby town of Siletz is mainly received as rain during fall and winter (November-March). The selected low gradient fish bearing reaches had minimal (LW) pieces present and were located in different tributaries: Site 1 is located in the main stem of Mill Creek, Sites 2 is located in Cerine Creek, and Site 3 is located in the South Fork (Table 1). All sites display low to moderately developed pool riffle sequences with bankfull discharge ($Q_{bf}$) between 2.4 and 8.7 m$^3$/s (Table 1).

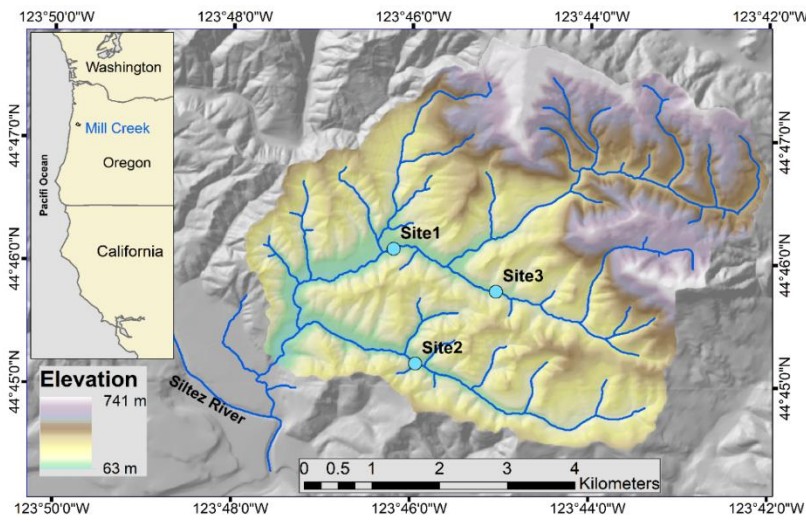

**Figure 1**. Location of the Mill Creek watershed, OR and the Study Sites 1, 2, and 3.

**Table 1**. Characteristics of the three Study Sites. Values in parenthesis correpsond to the standard errors

| Characteristic | Units | Site 1 | Site 2 | Site 3 |
|---|---|---|---|---|
| Drainage area | km$^2$ | 16 | 5 | 5 |
| Length | m | 119 | 123 | 115 |
| Bankfull discharge ($Q_{bf}$) | m$^3$/s | 8.7 | 2.4 | 2.5 |
| Bankfull width | m | 10.6 (1.9) | 5.5 (0.8) | 7.4 (1.6) |
| Bankfull depth | m | 0.7 (0.2) | 0.6 (0.1) | 0.6 (0.1) |
| Bankfull cross-sectional area | m$^2$ | 6.9 (1.8) | 3.3 (0.7) | 4.1 (0.9) |
| Slope | m/m | 0.0032 | 0.004 | 0.008 |
| D$_{50}$ | m | 0.039 | 0.0153 | 0.0297 |
| $\tau_c$ (Mueller et al., 2005). | N/m$^2$ | 16.1 | 6.7 | 16.8 |

5 **2.2. Field Methods**

During July of 2015, a detailed topographic survey was conducted in each of the three study reaches including 20–28 cross-sections (XS) per site spaced ~ ½ bankfull width apart and 1,700–2,000 additional survey points to characterize abrupt topographic changes. The raw topography was smoothed and interpolated to a dense point cloud using a natural neighbour scheme under ArcGIS and used in the model framework (see section 3.2). We estimated the grain size distribution (GSD) in

10 each reach based on particle counts (Wolman, 1954) conducted in 11–25 visually identified patches of relative uniform sediment size per site (Buffington and Montgomery, 1999b; Rosenberger and Dunham, 2005; Smith and Prestegaard, 2005; Cienciala and Hassan, 2013). We instrumented the study reaches with pressure transducers at a relatively stable and uniform XS (Fig. 2). Discharge was measured using the velocity-area method (Dingman, 2002) using a Hack FH950 Portable Velocity meter and depth-discharge rating curves were developed based on 9–10 discharge measurements per site covering a wide range

15 in discharge levels: 5-100% of $Q_{bf}$ in Site 1, 5–63% of $Q_{bf}$ in Site 2, and 5–89% of $Q_{bf}$ in Site 3.

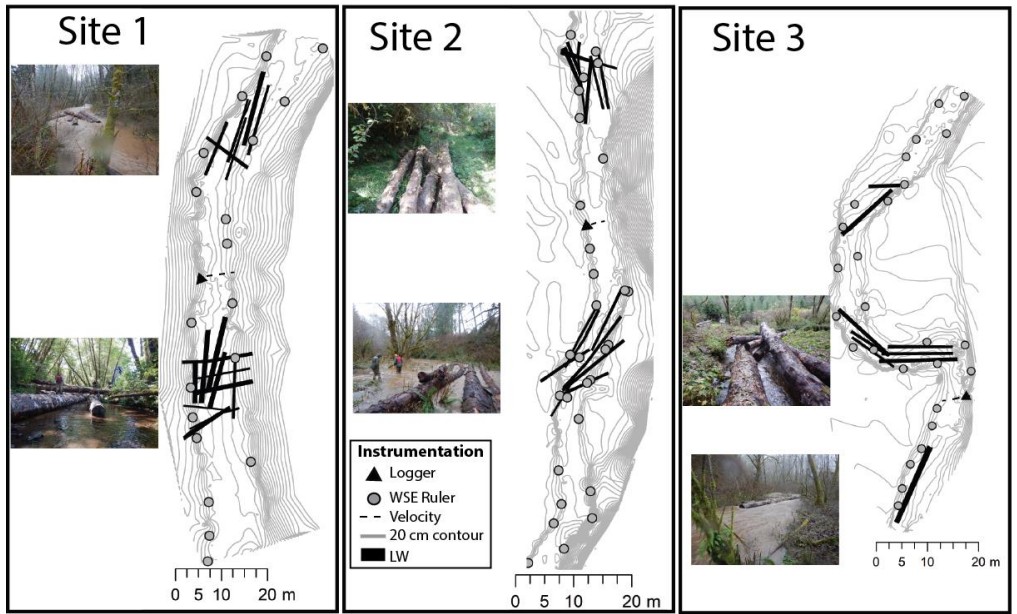

**Figure 2**. Topography (derived 0.2 m contours) and location of introduced large wood (LW), water surface elevation (WSE) monitoring rulers (circles), water level loggers (triangle), and location of velocity measurements (dash line). Flow direction is from top to bottom in all three sites.

In August of 2015, 39 pieces of LW were added to the three sites by the Oregon Department of Fish and Wildlife. The wood was arranged into two jams per site with 3–8 wood pieces each (Fig. 2). The wood pieces added were all over 6 m long with diameters between 0.5 m and 1.6 m. The logs were oriented lengthwise in the stream to mimic wood pieces that have been rafted into a location and provide the most contact with the bed and a stable but natural configuration to drive geomorphic change. The jams were located in bends in the stream reaches where possible, and additional logs were placed on top of jams and braced by existing trees to increase stability, but no other means of permanently fixing the jam locations was used. The entire process of building the six jams across sites took less than two days.

### 2.3. Flow modelling

In order to describe flow field changes triggered by the addition of LW, we used the 2D unsteady Nays2DH model (Takebayashi et al., 2003; Jang and Shimizu, 2005; Shimizu and Takebayashi, 2014; Nelson et al., 2016). This model was selected for its ability to simulate unsteady conditions experienced during rapidly varying discharge and rapidly varying shear stress around obstacles. We first simulated steady intermediate (20–50% of $Q_{bf}$) and large ($Q_{bf}$) flow levels for calibration purposes, and 35–45-hour long $Q_{bf}$ flow events (unsteady) before and after the LW additions. These unsteady models were used to characterize the distributions of depth, velocity, and shear stress pre- and post-LW addition (Table 2) and include a wide range of flows between 0.1 $Q_{bf}$ and $Q_{bf}$. The model uses a free surface, finite differenced, and depth integrated version of

the Navier-Stokes equations (NSE) assuming a logarithmic velocity profile in the boundary layer near the bed and a parabolic velocity profile away from it. Nays2DH uses the Cubic-Interpolated Pseudo-Particle (CIP) method for finite differencing which gives high accuracy flow predictions, particularly in instances of flow separating shear layers. The model calculates present and future 2D-velocity for a given time step using a cubic profile to determine its spatial derivative under the assumption that

both time steps follow the governing NSE flow equation (Yabe et al., 1990). This method requires the use of a short modelling time step to ensure model stability, thereby limiting the length of model runs given computational cost (Shimizu and Takebayashi, 2014; Nelson et al., 2016).

**Table 2.** Nays2DH model parameters and calibration results for pre and post large wood (LW) models for different discharge ($Q_i$) levels. Fractional bankfull discharge ($Q_i/Q_{bf}$); average model depth ($H_{mean}$); number of water surface
elevation (WSE) and velocity (U) observations taken, root mean squared error (RMSE), $R^2$ for WSE and time average U are indicated for calibration runs when available along with observed and modeled WSE slopes.

| Site | LW | $Q_i$ (m³/s) | $Q_i$ /$Q_{bf}$ | # Nodes | $H_{mean}$ (m) | # WSE obs. | # U obs. | RMSE-WSE (m) | RMSE-U (m/s) | $R^2$ WSE | $R^2$ U | WSE Slope obs. (%) | WSE Slope Modeled (%) |
|---|---|---|---|---|---|---|---|---|---|---|---|---|---|
| 1[a] | pre | 4.53 | 0.52 | 60621 | 0.55 | 15 | 24 | 0.025 | 0.34 | 0.97 | 0.41 | 0.29 | 0.29 |
| 1[a] | pre | 8.7 | 1.00 | 60621 | 0.71 | 15 | - | 0.032 | NA | 0.94 | NA | 0.35 | 0.37 |
| 1[a] | post | 1.91 | 0.22 | 60621 | 0.57 | 13 | 19 | 0.078 | 0.16 | 0.94 | 0.39 | 0.75 | 0.76 |
| 1[b] | post | 12 | 1.38 | 60621 | 0.98 | 6 | - | 0.212 | NA | 0.66 | NA | 0.86 | 0.89 |
| 2[a] | pre | 1 | 0.41 | 58176 | 0.421 | 24 | 13 | 0.025 | 0.12 | 0.97 | 0.87 | 0.42 | 0.41 |
| 2[a] | pre | 2.43 | 1.00 | 58176 | 0.56 | 24 | 16 | 0.026 | 0.26 | 0.97 | 0.87 | 0.39 | 0.43 |
| 2[a] | post | 1.49 | 0.61 | 58176 | 0.53 | 25 | 20 | 0.034 | 0.26 | 0.98 | 0.68 | 0.66 | 0.68 |
| 2[b] | post | 3.8 | 1.56 | 58176 | 0.58 | 10 | - | 0.085 | NA | 0.90 | NA | 0.51 | 0.47 |
| 3[a] | pre | 1.08 | 0.49 | 53169 | 0.34 | 26 | 17 | 0.045 | 0.26 | 0.98 | 0.70 | 0.88 | 0.86 |
| 3[a] | pre | 2.2 | 1.00 | 53169 | 0.46 | 22 | 24 | 0.023 | 0.36 | 1.00 | 0.70 | 0.87 | 0.85 |
| 3[a] | post | 1.09 | 0.50 | 53169 | 0.55 | 24 | 24 | 0.036 | 0.26 | 0.99 | 0.69 | 1.16 | 1.15 |
| 3[b] | post | 3.5 | 1.59 | 53169 | 0.58 | 11 | - | 0.134 | NA | 0.96 | NA | 1.12 | 1.17 |

[a] assuming a constant downstream water surface elevation as the initial boundary condition
[b] uniform flow assumption as the initial upstream and downstream boundary condition

15       Model input data were channel topography, discharge, roughness, downstream flow stage, and a characterization of the initial upstream WSE condition. Given the large size of the LW pieces with diameters 0.8–2.3 times the $Q_{bf}$ depth in all three sites, they were represented in the model as fully penetrating the water depth protruding into the channel and located based on detailed topographic surveys. Based on time-lapse photography and flow level observations, the LW pieces were never overtopped by the flow. In cases where LW pieces were angled relative to the slope of the streambed or where lateral
topography in the bed left large gaps under LW pieces, the shape of the flow restricting obstacles were adjusted to allow for a significant amount of flow to pass around the structures.

        The model parameters were adjusted based on 1,000 seconds constant discharge calibration simulations with 0.01-second time steps averaged over 10 iterations. We assumed a constant downstream WSE measured in the field for all calibration runs except for the high flows modelled after LW addition when wading was hazardous. For these runs, and for the
hydrograph simulations, we employed the uniform flow assumption as the initial upstream and downstream boundary condition (Table 2). The model equations for downstream (*u*) and cross-stream (*w*) velocity components are solved over an orthogonal,

curvilinear grid (Shimizu and Takebayashi, 2014; Nelson et al., 2016) and used to estimate the shear stress ($\tau$) via a unitless coefficient of bed shear force ($C_f$):

$$\tau = \rho C_f (u^2 + w^2) \tag{1}$$

where $\rho$ is water density and $C_f$ is estimated based on a spatially variable unitless Manning roughness coefficient ($n$) calculated for the identified sediment patches based on the grain size ($D$), gravitational acceleration ($g$), flow depth ($h$) and an unitless $\alpha$ parameter that can vary from 1 to 3:

$$n = \frac{(\alpha D)^{1/6}}{7.66\sqrt{g}} \tag{2}$$

$$C_f = \frac{n^2 g}{h^{1/3}} \tag{3}$$

Roughness values for vegetated areas outside the channel were set to be 10% higher than the maximum patch $n$ value in each model. The best fits for all three sites were found with $\alpha = 3$ and $D=D_{84}$ (size of a particle equivalent to the 84[th] percentile in a cumulative frequency distribution). We chose to model turbulence using the zero-equation option in the model, which assumes smooth changes in lateral topography, and thus $\tau$ and $h$ dominate the momentum transport. A spatially varying eddy viscosity is calculated in the model as a ratio of the depth and velocity.

We calibrated the models by comparing observed and predicted WSE through each reach, with and without LW, and iteratively adjusting $C_f$ by changing $n$. The root mean squared error for the WSE, computed based on 6–26 observations per flow, was below 0.045 m for all pre-wood scenarios and no more than 0.21 m for all post wood models (Table 2). Abrupt changes to streambed morphology after the addition of LW contributed to model error, as these changes could alter the observed WSE but were not reflected in our models. For example, on the downstream end of Site 1 we observed significant sediment deposition on the right side of the channel and scour on the left side. Aside from this, the model was able to accurately capture the large changes in WSE across log-jams and the general water surface slope (Table 2). Velocity observations were used as an additional check after calibration for 2–3 flow conditions per site when wading was possible. The RMSE of velocity varied between 0.11–0.36 m/s (Table 2) based on 13–24 observations taken across the streams (Fig. 2). These values are similar to other reported values of model RMSE for WSE and velocity for efforts that did not include wood; indicating overall strong performance of the model (Cienciala and Hassan, 2013; Mueller and Pitlick, 2014; Segura and Pitlick, 2015a; Katz et al., 2018).

## 2.4 Data analysis

We evaluated the changes in velocity and shear stress triggered by the addition of LW in the three study reaches during a $Q_{bf}$ flow event with emphasis on the peak discharge. Then we quantified the differences in the spatial extent of suitable habitat for juvenile Coho Salmon during bankfull flow and during the duration of a complete hydrograph in which discharge varied between $0.1Q_{bf}$ and $Q_{bf}$. For both velocity and shear stress distributions, only areas where depth > 0.1 m and velocity or shear stress > 0.01 units were included to limit the study to the active channel and depths where model assumptions were not

likely to be violated and to areas of the channel in which juvenile fish were likely to be found (Bustard and Narver, 1975b). We estimated the area with acceptable fish habitat within the modelled domains using a critical swimming velocity ($v_{crit}$) of 0.5 m/s and a burst swim velocity ($v_{burst}$) of 1 m/s for winter-time juvenile Coho Salmon (Glova and McInerney, 1977; Taylor and McPhail, 1985). The $v_{crit}$ corresponds to the maximum velocity at which a fish can maintain position in the flow field for extended periods at a specific temperature and $v_{burst}$ represents a maximum instantaneous swim velocity.

Since juvenile salmonids are often shelter in substrate during harsh environmental conditions (Hartman, 1965; Rimmer et al., 1983; Bradford et al., 1995; Cunjak, 1996; Bradford and Higgins, 2001), we used the predicted shear stress values to estimate the proportion of the bed area in which the entrainment of the $D_{50}$ is likely. Indeed, the $D_{50}$ values in the study sites range from 16–39 mm which is similar to the particle size in which sheltering juvenile Atlantic salmon have been observed (Cunjak, 1988). Our assumption is that transport of the $D_{50}$ is a reasonable threshold to represent conditions in which dislodging fish is possible because the substrate would fail to provide shelter. The critical shear stress ($\tau_c$) associated with the movement of the $D_{50}$ was estimated based on slope ($s$) (Mueller et al., 2005):

$$\tau_c^* = 2.18s + 0.021 \tag{4}$$

$$\tau_c^* = \frac{\tau_c}{(\rho_s - \rho)g D_{50}} \tag{5}$$

where $\tau_c^*$ is the dimensionless critical Shield's stress and $\rho_s$ is sediment density (i.e., 2,500 kg/m$^3$ for sandstone). We assumed that channel bed locations with $\tau > 2\tau_c$ are likely to experience full transport mobility (Wilcock and McArdell, 1993), and therefore offer no fish sheltering given that most of the available particles sizes would likely be mobilized. In sections of the bed experiencing partial transport ($\tau_c < \tau < 2 \tau_c$) we assumed that sheltering would be difficult but not impossible as larger particles are likely to remain stable.

## 3 Results

### 3.1 Comparison of velocity before and after the addition of LW

According to model predictions the mean bankfull flow velocity ($v$) before LW additions ranged between 0.7 m/s and 1.2 m/s while after the addition of LW pieces velocity ranged between 0.53 m/s and 0.92 m/s (Table 3), corresponding to 23.2–36.3% decreases. The distributions of velocity values at wetted points throughout the model domain were narrower before LW was added than the distributions after the LW (Fig. 3). Before the restoration, all velocity distributions were relatively homogenous with high density of observations around the mean values (Fig. 4) and relatively small standard deviations (0.3 m/s to 0.5 m/s, Table 3). After the LW additions, the flow fields became more heterogeneous (standard deviations between 0.4 m/s and 0.7 m/s, Table 3) with lower clustering of velocity values around the mean and a greater proportion of areas in the channel bed that experienced extreme (low and high) velocity conditions (Fig. 3 and 4). The increased heterogeneity of flow conditions after the LW additions was associated with a greater proportion of flow interacting with the floodplains upstream of the LW jams and the flow passing through the decreased cross-sectional area of the LW jams themselves. The decrease

flow area around the wood is consistent with the increase in the mean WSE slope between pre- and post-LW in all sites (Table 2).

**Table 3.** Mean and standard deviation (SD) of velocity ($v$) and shear stress ($\tau$) at bankfull flow ($Q_{bf}$) pre- and post-LW at the three study sites; $Q_{bf}$ modelling results for habitat metrics $v$ and $\tau$ expressed as a percentage of the channel bed pre- and post-LW; and percentage change in available fish habitat.

| Metric | Site 1 | | Site 2 | | Site 3 | |
|---|---|---|---|---|---|---|
| | Pre-LW | Post-LW | Pre-LW | Post-LW | Pre-LW | Post-LW |
| Mean $v$ (SD) | 1.23 (0.5) | 0.92 (0.7) | 0.69 (0.3) | 0.53 (0.4) | 1.02 (0.5) | 0.65 (0.5) |
| Mean $\tau$ (SD) | 23.41 (14.7) | 18.99 (29.7) | 12.24 (8) | 9.14 (11.8) | 22.27 (18.2) | 11.35 (16.8) |
| **% of Bed** | | | | | | |
| $v \leq v_{crit}$ [a] | 15.3 | 32.5 | 26.7 | 52.2 | 23.8 | 47.3 |
| $v_{crit} \leq v \leq v_{burst}$ [a] | 13.9 | 32.7 | 60.2 | 34.5 | 17.0 | 28.3 |
| $v > v_{burst}$ [a] | 70.8 | 34.8 | 13.1 | 13.3 | 59.2 | 24.4 |
| $\tau < \tau_c$ [b] | 30.2 | 69.0 | 29.1 | 59.9 | 41.0 | 76.4 |
| $\tau_c < \tau < 2\tau_c$ [b] | 43.9 | 12.6 | 23.4 | 17.9 | 35.3 | 14.8 |
| $\tau < 2\tau_c$ [b] | 25.8 | 18.4 | 47.5 | 22.2 | 23.7 | 8.9 |
| **% Change in Available Habitat** | | | | | | |
| $v \leq v_{crit}$ [a] | +112.8% | | +95% | | +99.3% | |
| $v_{crit} \leq v \leq v_{burst}$ [a] | +134.5% | | -42.6% | | +66.1% | |
| $v > v_{burst}$ [a] | -50.8% | | +1.4% | | -58.8% | |
| $\tau < \tau_c$ [b] | +128.3% | | +105.9% | | +86.3% | |
| $\tau_c < \tau < 2\tau_c$ [b] | -71.4% | | -23.5% | | -58.2% | |
| $\tau < 2\tau_c$ [b] | -28.8% | | -53.3% | | -62.6% | |

[a] $v_{crit}$ is 0.5 m/s and $v_{burst}$ = 1 m/s

[b] $\tau_c$ is the critical shear stress for the movement of the median grain size (Table 1)

The predicted reduced velocity in the stream channels after the addition of LW indicated increased fish habitat in all sites. The proportion of the wetted channel area with velocity values below the critical ($v \leq v_{crit}$) increased over 95% in all sites (Table 3, Fig. 3) being highest in Site 1. The absolute increases in the total area where $v \leq v_{crit}$ were even greater at 186.1%, 141.2%, and 169.5% for sites 1–3 respectively. These values may be more relevant to restoration success in the context of density dependent habitat limitations faced by juvenile Coho Salmon. The LW pieces backed up flow, increasing the wetted width, which resulted in additional low velocity habitat created beyond the original channel margins (Fig. 3). Hence, the wetted areas of sites 1, 2 and 3 increased by 34%, 22%, and 35% respectively (Fig. 3). The areas with temporarily acceptable habitat ($v_{cirt} \leq v \leq v_{burst}$) also increased by 134.5% in Site 1 and by 66.1% in Site 3 of their wetted channel area (Table 3). Conversely, temporarily acceptable habitat decreased from 60.2% to 34.5% of the wetted bed in Site 2 (Table 3). This site had proportionally more areas with $v < v_{busrt}$ prior to the LW introductions (light blue in Fig. 3) and therefore less potential for an increase in that category. These predictions clearly indicate that the LW additions increased the area of habitat acceptable for juvenile salmon at $Q_{bf}$.

As mentioned above, the velocity distributions changed in shape with the highest frequency values shifting away from the value of $v_{burst}$ to below or near the value of $v_{crit}$, hence the skewness of the distributions shifted from negative to

positive values in all sites. This shift provides assurance of the robustness of our results. If the thresholds used to determine habitat acceptability were shifted slightly, to account for variations in other habitat parameters such as water temperature or fish size, the benefits predicted by our model results would remain consistent.

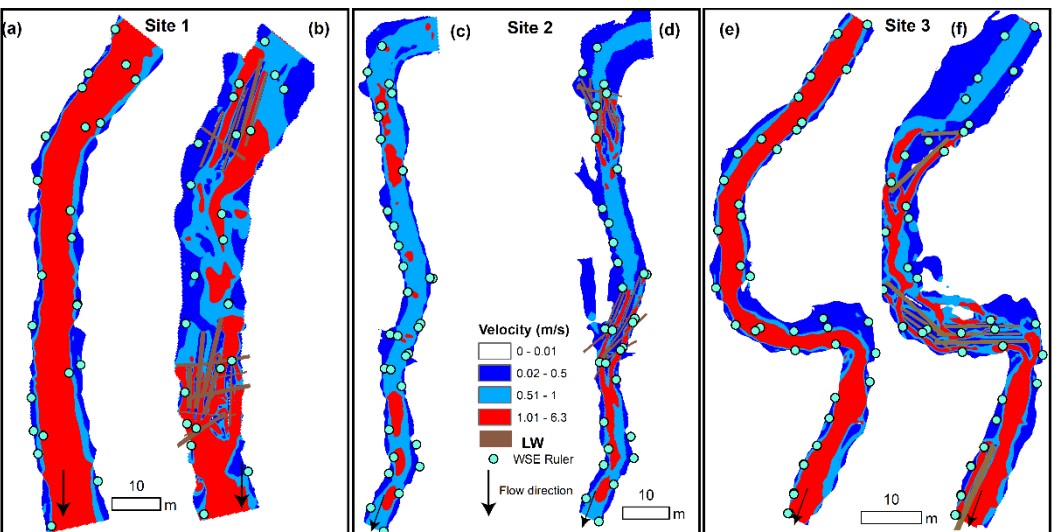

5    **Figure 3**. Mean flow velocity at bankfull discharge before (a, c, and e) and after (b, d, and f) the addition of large wood (LW) in Sites 1, 2, and 3. The colors correspond to thresholds of velocity relevant to the ability of juvenile Coho Salmon to maintain position in the stream: dark blue means $v < v_{crit}$ where $v_{crit} = 0.5$ m/s, light blue means $v_{crit} < v < v_{burst}$ where $v_{burst} = 1$ m/s, and red means $v > v_{burst}$. The location of the installed water surface rulers is included to facilitate visual comparison of the increase extend of floodplain inundation in each site during bankfull conditions.

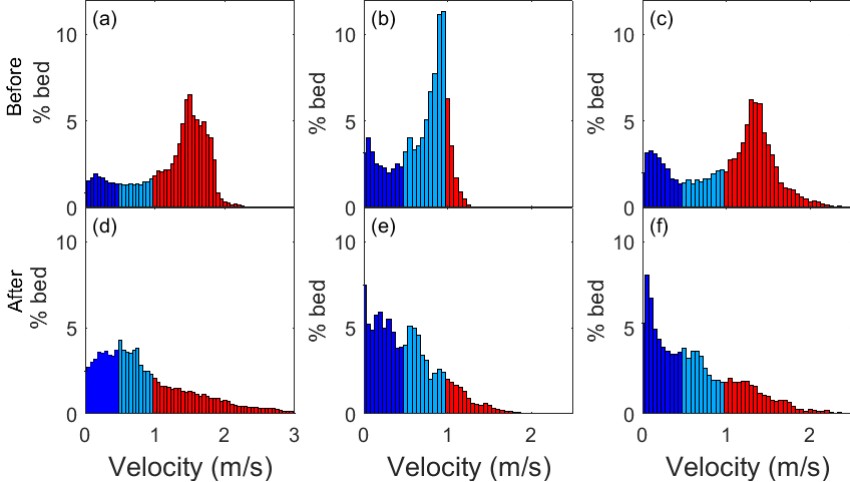

**Figure 4.** Velocity distributions at bankfull flow at Sites 1 (a, d), 2 (b, e) and 3 (c, f) before (a-c) and after (d-f) the addition of large wood (LW). The colors correspond to thresholds of velocity relevant to the ability of juvenile Coho Salmon to maintain position in the stream: dark blue means $v < v_{crit}$ where $v_{crit} = 0.5$ m/s, light blue means $v_{crit} < v < v_{burst}$ where $v_{burst} = 1$ m/s, and red means $v > v_{burst}$.

**3.2 Comparison of shear stress before and after the addition of LW**

Model predictions indicated that the reach-average $Q_{bf}$ values of shear stress ($\tau$) before the LW additions were 23.41 N/m$^2$ in Site 1, 12.24 N/m$^2$ in Site 2, and 22.27 N/m$^2$ in Site 3 (Table 3). Modelling results indicated 18–49% reductions in shear stress after the LW pieces were added, which resulted in substantial increase of fish habitat in terms of substrate stability. Considering the critical Shields value for the median grain size (Table 1), the proportions of the wetted bed with stable conditions ($\tau < \tau_c$) increased from 29–41% before LW to 59.9–76.4% after wood was added—an overall increase in fish habitat of 86–128% (Table 3). Further, the total increases in absolute area where $\tau < \tau_c$ were 205.8% for Site 1, 151.4% for Site 2, and 151.6% for Site 3 (Fig. 5). The spatial changes in the distributions of shear stress were associated with consistent decreases in flow velocity near the channel margins and additional stream connectivity with available floodplains (Fig. 5). Additionally, increased WSE slope through the reaches after the addition of LW helped drive the variation in shear stress through the formation of deeper pools upstream of LW jams.

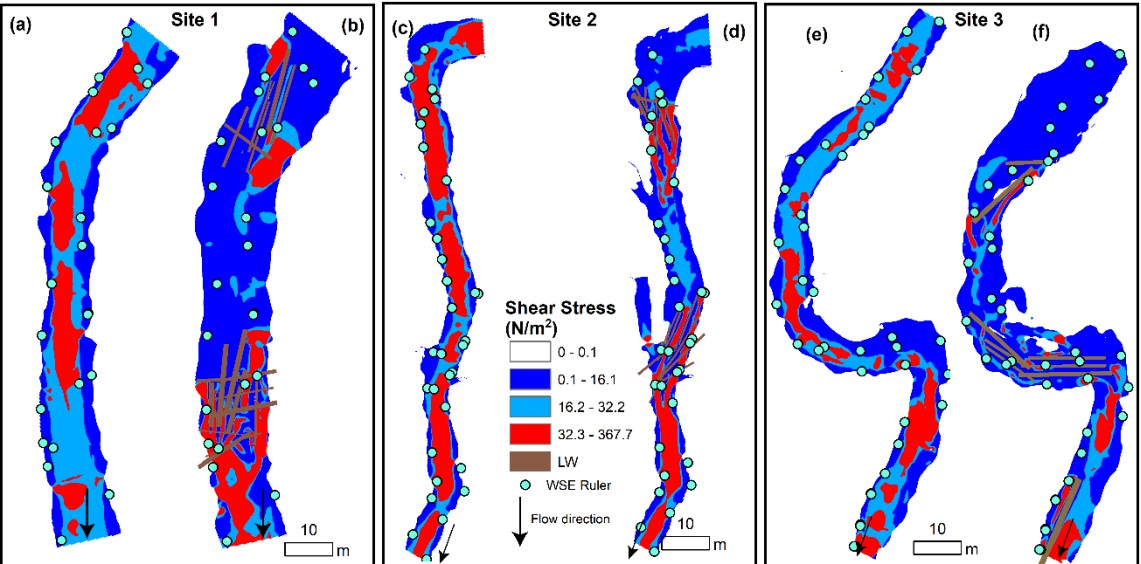

**Figure 5.** Spatial distributions of shear stress ($\tau$) at bankfull discharge before (a, c, and e) and after (b, d, and f) the addition of large wood (LW). Dark blue corresponds to $\tau < \tau_c$, light blue corresponds to $\tau_c < \tau < 2\tau_c$, and red corresponds to $\tau > 2\tau_c$. The location of the installed water surface rulers is included to facilitate visual comparison of the increase extend of floodplain inundation in each site during bankfull conditions.

The shape of the distributions of shear stress changed, from having a distinct peak near the mean in addition to a high frequency of observations near zero, to a distribution characterized by a constant decay (Fig. 6). We fitted the mean normalized distributions of shear stress before and after the LW additions to a gamma function (Segura and Pitlick, 2015b) and found that the shape parameter ($\alpha$) of the distributions decreased for all sites. While this parameter before LW varied between 2.2 and 5.4 it varied between 0.6 and 1.0 after the LW additions. These changes illustrate increases in complexity in the flow field after the restoration project.

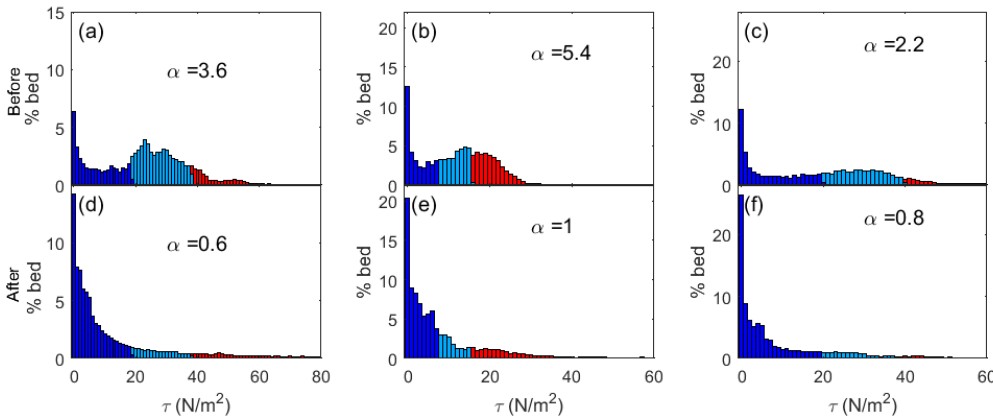

**Figure 6.** Shear Stress ($\tau$) distributions at bankfull flow at sites 1 (a, d), 2 (b, e) and 3 (c, f) before (a-c) and after (d-f) the addition of large wood (LW). Alpha ($\alpha$) paramters of the a gamma fit are provided. Dark blue corresponds to $\tau < \tau_c$, light blue corresponds to $\tau_c < \tau < 2\tau_c$, and red corresponds to $\tau > 2\tau_c$.

## 3.3. Temporal variability in available habitat during full bankfull flow events

Modelled results before LW additions during the entire hydrographs indicated that the reach area with acceptable habitat in terms of velocity ($v < v_{crit}$) varied between 15% and 36% in Site 1, 27 % and 74% in Site 2, and 23% and 38% in Site 3 (Table 4, Fig. 7 a–c). These percentages of reach area with acceptable habitat increased after the addition of LW to 31–74% in Site 1, 48–85% in Site 2, and 42–72% in Site 3 (Fig. 7 a–c) indicating average increases between 23 % and 29 % (Table 4 and Fig. 7 a–c). The temporal variability in the percentage of the channel with acceptable habitat ($v < v_{crit}$) reflects differences in flood plain connectivity among sites. For instance, the consistent increase in acceptable habitat ($v < v_{crit}$) area in sites 2 and 3 over the duration of the entire hydrograph (Fig. 7 e–f) is likely a result of their large available floodplain area (Fig. 2). Conversely, Site 1 experienced a wider range of increases in acceptable area after LW addition with the smallest differences occurring around the peak discharge (Fig. 7 d). This is likely the result of water completely inundating the site's relatively smaller available floodplain area (Fig. 2) during the rising limb of the hydrograph.

Similar to the modelling results for velocity the proportion of the wetted channel with acceptable habitat for fish to shelter within channel bed sediment increased for all flow levels during the entire hydrograph simulations in all study sites (Fig. 7 a–c). The percentage of the channel bed with stable substrate ($\tau < \tau_c$) before LW varied between 30% and 92% in Site 1, between 28% and 70% in Site 2 and between 41% and 79% in Site 3. These ranges increase on average 27-30% to 68–93 % in Site 1, 57–82% in Site 2 and 76–94 % in Site 3 (Fig. 7 a–c). Unlike what was observed for velocity, there were significant temporal variations in the proportion of the wetted channel with stable substrate ($\tau < \tau_c$), especially at Site 1 (Fig. 7d) which experienced the widest range of change between -2% and 42% (Table 3). In this site, the greatest increase in relative habitat area occurred at the peak of the hydrograph when presumably conditions would be the harshest for juvenile Coho Salmon (Fig. 7 d). In other words, the greatest increases of proportional area with stable substrate after LW addition coincided with the high discharge, while smaller differences took place at the initial low discharge values. In addition, larger immobile substrate areas

were evident during the falling limb of the hydrograph compared to the rising limbs in all sites (Fig. 7 d–f). This is likely associated with temporary storage of water in the floodplain after the addition of LW and a related decreased transport capacity (decreased shear stress) available to mobilize bed material.

**Table 4.** Hydrograph modelling results for habitat metrics of velocity (v $\leq$ $v_{crit}$[a]) and shear stress ($\tau$ < $\tau_c$[b]) expressed as the range of the percentage of the channel bed pre- and post-LW; and change in percentage available as fish habitat pre- and post-LW

| % of Bed | Site 1 | | Site 2 | | Site 3 | |
|---|---|---|---|---|---|---|
| | Pre-LW | Post-LW | Pre-LW | Post-LW | Pre-LW | Post-LW |
| v $\leq$ $v_{crit}$[a] | 15–36 | 31–74 | 27–74 | 48–85 | 23–38 | 42–72 |
| change in % | 16–42 (29) | | 12–27 (23) | | 24–35 (29) | |
| $\tau$ < $\tau_c$[b] | 30–92 | 68–93 | 28–70 | 57–82 | 41–79 | 76–94 |
| change in % | -2–42 (28) | | 13–32 (27) | | 16–36 (30) | |

[a]$v_{crit}$ is 0.5 m/s

[b]$\tau_c$ is the critical shear stress for the movement of the median grain size (Table 1)

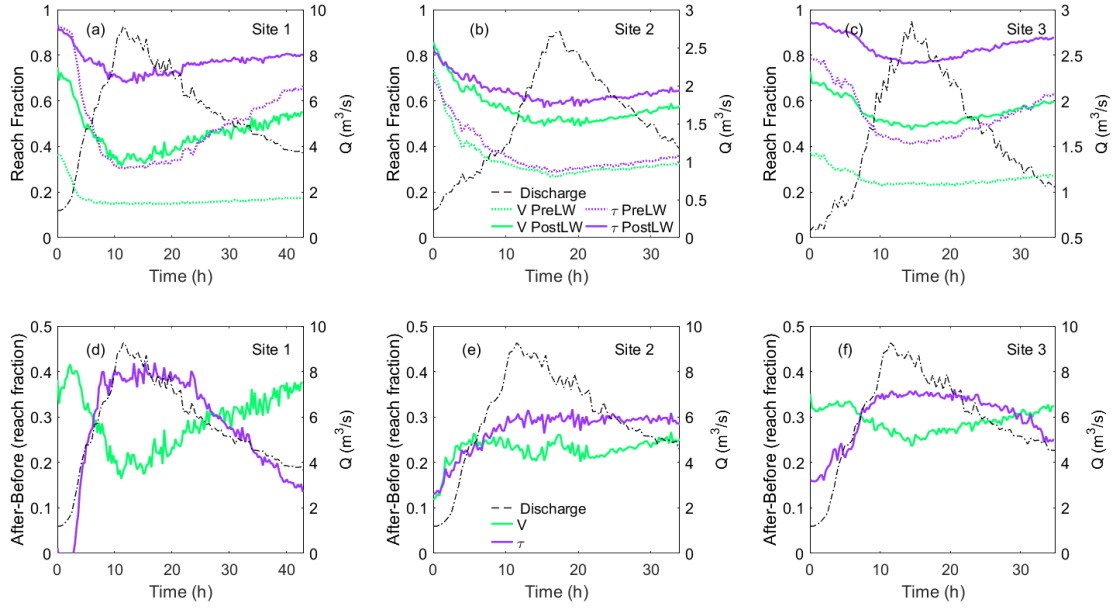

**Figure 7:** a–c: Fraction of the flow domain with $v$ < $v_{crit}$ or $\tau$ < $\tau_c$ during simulated 40–35 hour bankfull flow events in the 3 study sites pre- and post LW; d–f: differences between after and before LW additions in the fraction of the reach area with $v$ < $v_{crit}$ or $\tau$ < $\tau_c$.

## 4 Discussion

The goal of this study was to model the hydraulic effects of the introduction of LW on components of fish habitat in three gravel bed streams. Two-dimensional (2D) modelling predicted significant changes in the flow field pre- and post-LW

additions that resulted in approximately twice as much simulated winter rearing habitat in all sites. To our knowledge, this study is the first to simulate the impact of the addition of LW on fish habitat at the reach scale using a field calibrated, unsteady 2D hydraulic model, calibrated to pre and post LW flow events. Our findings concur with uncalibrated and steady state simulations that have documented increases in the heterogeneity in the flow field at high discharges after the addition of LW,

thereby increasing fish habitat (He et al., 2009; Hafs et al., 2014; Wall et al., 2016). The use of water surface elevation and velocity calibration data in pre- and post-LW models provided a robust framework to estimate mean depth-averaged flow velocity and shear stress, variables likely to fully represent realistic winter sheltering opportunities for juvenile fish in terms of flow velocity and substrate stability.

        The addition of LW in the study reaches modified river hydraulics, resulting in significantly wider wetted areas. At

bankfull flow, the increased floodplain connectivity was associated with more heterogeneous flow fields characterized by wider distributions of velocity and shear stress with overall lower mean values. The shapes of pre-LW velocity distributions for all sites were similar to those observed in small mountain streams with large frequency at both intermediate and low velocity values (Cienciala and Hassan, 2016). The shape of the velocity distributions changed dramatically post-LW being characterized by a higher proportion of low velocity areas in all three sites. Other field modelling efforts have documented

similar effects of LW on the velocity distribution in the flow field (Wall et al., 2016). Flume experiments as well as field simulations have also reported reductions in flow velocity with increasing large wood obstacles (He et al., 2009; Davidson and Eaton, 2013; Hafs et al., 2014). The distributions of shear stress also changed dramatically from closely resembling those observed in single thread streams pre-LW (Lisle et al., 2000; Mueller and Pitlick, 2014; Segura and Pitlick, 2015a; Cienciala and Hassan, 2016) to resembling complex braided channels (Paola, 1996; Nicholas, 2003; Mueller and Pitlick, 2014;

Tamminga et al., 2015) post-LW. The shift towards a greater frequency of low shear stress is likely attributed to shear stress partition by the channel banks and LW form drag (Kean and Smith, 2006; Yager et al., 2007; Ferguson, 2012; Scheingross et al., 2013). The changes in the velocity and shear stress distributions occurred as the flow encroached into the floodplain, and although stream margins have been associated with the creation of off-channel habitat for juvenile Coho Salmon in previous studies (Swales and Levings, 1989; Bell et al., 2001), no quantification of the actual changes in the flow field in terms of

velocity or shear stress had been conducted before. The post-LW distributions of shear stress and velocity indicated increased hydraulic and habitat heterogeneity (Gerhard and Reich, 2000; Brooks et al., 2006), which has been reported as a key flow field characteristic associated for habitat suitability for salmonids (McMahon and Hartman, 1989; Roni and Quinn, 2001; Venter et al., 2008; Anlauf-Dunn et al., 2014). The suggested benefits of flow heterogeneity include velocity refuges in close proximity to feeding locations and cover from predators (Nickelson et al., 1992a; Nickelson and Lawson, 1998; Gustafsson et

al., 2012). The increase availability of low velocity areas during bankfull discharge is relevant for winter fish habitat given the high mortality that can occur during this season (Quinn and Peterson, 1996). Although we did not measure sediment transport, the overall reduction of velocity and shear stress likely contributes to increased pool depth and area (Montgomery et al., 1995; Beechie and Sibley, 1997; Collins et al., 2002) and decreased overall bed load transport capacity (Thompson and Fixler, 2017; Wohl and Scott, 2017).

Although we were able to model velocity and shear stress, there are components of the flow and temporal changes to the bed that we were unable to account for. While the sharp topography in our model domains around LW pieces allowed us to predict local areas of elevated shear stress, the 2D model is not capable of capturing the strong vertical currents that are likely to develop in proximity to the LW and deform the stream bed with important impacts on the assessment of available

habitat (Mutz et al., 2007). While it has been observed that 3D models outperform 2D models in predicting flow structures in close proximity to obstacles (Shen and Diplas, 2008), our results are promising. The full depth penetrating size and downstream orientation of the LW pieces in our reaches resulted in predictions of fragmented flow, increased maximum local shear values, deflection of maximum velocities and shear stress away from the outside of bends, and low velocity habitat regions in the wake of longitudinally oriented logs which align with observations made in other studies using 3D modelling (Daniels and

Rhoads, 2003, 2004b, a; Xu and Liu, 2017). Despite these promising observations, there still remains uncertainty around the 3D nature of the flow, which is likely greater in areas with denser LW loading and greater stream curvature and during periods of increased discharge (Daniels and Rhoads, 2004a). A comparison of observed and modelled velocity values near LW structures would provide further understanding of the uncertainty of our 2D modelling approach and insight into the accuracy of the predictions. A comprehensive set of measurements could show a potential envelope around complex LW structures

where model predictions are less accurate. However, this was not possible in our case given logistical constrains to collect such data. A 3D version of the NAYS2DH model, known as NaysCUBE, could potentially address some of these issues. However, this approach would require substantially more time, computational power, and calibration to ensure model stability. As the bed deforms, we would expect to see a feedback of changing velocity and shear stress values, particularly where we predicted the highest values. Another limitation of our approach is the inability to account for LW mobility. Field observations

during high flows and the length of LW pieces relative to stream widths, indicate that pieces were unlikely to mobilize downstream (Merten et al., 2010; Ruiz-Villanueva et al., 2016); however, we did observe some floating and minor adjustment of some LW pieces (particularly in Site 1) during the highest flow events. Thus localized stream hydraulics could be subject to variations (Daniels and Rhoads, 2004a), including potential flow underneath LW pieces, via both scour and hyporheic flow through sediment (Ruiz Villanueva et al., 2014) that we did not account for in the model. As the LW jams continue to develop

over many flow events, in addition to some movement of logs, smaller wood pieces, sticks, and leaves from the upper watersheds will also collect and have been shown to meaningfully alter flow through LW jam (Manners et al., 2007).

Despite these uncertainties, the strong agreement between observed and predicted water surface elevation and velocity before and after the LW additions provided evidence that the predictions are robust. This implies that this unsteady model, which has traditionally been used in larger systems (Kafle and Shakya, 2018) can be implemented in significantly smaller

systems even in the presence of large obstacles. Though it is key that sufficient detail on channel morphology in the regions where LW blocks flow is available to allow for conveyance through the model domain in such small streams.

The reach scale of this study should also be considered in viewing the results. Fully loading the watershed with LW at a similar density to our study sites may reduce the increase in WSE slope we observed after the addition of LW by backwatering areas where our downstream boundary conditions were located. This may lead to less heterogeneity of velocity

and shear stress in the flow field, particularly fewer values in the medium to high range. However, the changes in the flow field we documented clearly show that the addition of LW created more of the slow water habitat preferred by juvenile Coho Salmon during the winter. The observed shift in velocity distribution toward very low water velocities, particularly evident at Sites 2 and 3, may be especially important due to the energetic challenges faced by Coho Salmon in winter when food resources

and assimilation capabilities are limited (Cunjak, 1996; Huusko et al., 2007). Juvenile Coho Salmon are generally found in microhabitats with water velocities far below $v_{cr}$ in the winter (Bustard and Narver, 1975a; McMahon and Hartman, 1989), and the availability of such habitats, especially during high flows, may be a critical factor in increasing overwinter survival. The spatial arrangement of these low velocity habitats relative to water depth and cover in the form of woody debris and overhanging banks is also important, as these factors affect the risk of displacement and predation for juvenile Coho Salmon

(Bustard and Narver, 1975b; Tschaplinski and Hartman, 1983; McMahon and Hartman, 1989). Given the importance of multiple factors in winter habitat selection by Coho Salmon, incorporating velocity, depth, and cover into the habitat-modelling process would be a useful future direction for predicting the effects of LW addition on habitat suitability.

Considering that after the restoration project the flow field in the study reaches is adjusting to the new condition the model predictions will progressively lose accuracy as channel scouring and aggradation occur around and behind the new LW

additions. The period over which the predictions would be robust is uncertain and would depend on how fast the streams adjust to the new conditions and how stable the LW additions are. Both the stability of individual LW pieces and its function in the flow field depend of the size of the LW piece relative to the size of the stream. Modelling predictions indicated more habitat created in the large reach (Site 1) compared to the smaller reaches (Sites 2 and 3) both in terms of velocity and shear stress (Table 3). However, the introduced LW would likely be more stable in the smaller sites than in larger sites (Gurnell et al.,

2002; Hassan et al., 2005; Wohl and Jaeger, 2009; Merten et al., 2010; Ruiz-Villanueva et al., 2016) given not only difference in size (e.g., smaller sites being more narrow) but also differences in discharge. Therefore, we anticipate that the model predictions will lose accuracy sooner in the larger site and that there may be a trade-off between the timing and the resilience of restoration benefits. That is the addition of LW would likely increase the amount of suitable habitat sooner in the larger site but the LW pieces in this site also have the highest potential to leave the system. In order to test this expectation the model

could be run again with updated topography to explore how the predicted distributions of shear stress and velocity presented in this study compare to new estimations after the bed has adjusted. This would provide not only a way to contrast model predictions but to understand which site is changing faster after the restoration and what habitat benefits are likely to persist in the longer term (Wall et al., 2016). This trade-off relative to stream size and potential to LW export also highlight the importance of considering restoration in a basin wide context.

Although we focused on juvenile Coho Salmon in our analysis, the modelling results are highly relevant to other salmonid species in these streams, as well as to other life history stages. For example, the critical swimming speed of juvenile steelhead trout falls between the $v_{crit}$ and $v_{burst}$ values for Coho Salmon used in our analysis (Hawkins and Quinn, 1996), and so the amount of suitable habitat for juvenile steelhead following LW addition would also be expected to increase significantly. Furthermore, juvenile steelhead are more oriented to the stream bottom in winter than Coho Salmon, with age-0 steelhead

often using substrate as cover (Bustard and Narver, 1975a). As a result, the increased bed stability we observed post-LW would likely have an even stronger effect on habitat suitability for juvenile steelhead than for Coho Salmon. Changes in shear stress and bed stability can also have important effects on the survival of salmonid embryos incubating in the substrate (Lisle and Lewis, 1992), and our sites are located in important spawning areas for adult Coho Salmon and steelhead in the study basin.

More detailed examination of spawning sites, sediment transport, and scour depths would be needed to fully investigate effects of LW on salmonid embryo survival, but the modelling approach used here could provide valuable insight into the spatial distribution of shear stress in a study of this kind.

**Conclusions**

In this study, we used an unsteady two-dimensional hydraulic model to investigate the effects of the introduction of
large wood (LW) on fish habitat in three gravel bed streams. The models predicted habitat increases in terms of suitable flow velocity and area of stable substrate of over 80% in all streams. Our study is the first to use a field-calibrated model to estimate river hydraulics pre- and post-LW at the reach scale. The distributions of velocity and shear stress changed dramatically from bimodal to exponential decay, indicating increased flow complexity in the presence of LW and resembling a change from single thread to multithread channels. We observed larger changes in the largest site, however we anticipate a trade-off between
the timing and the resilience of restoration benefits given the higher likelihood for wood transport in the larger site. The methodology presented here can be used in the future as a tool to predict changes triggered by restoration efforts, evaluate long-term responses to restoration, and assess the changes in the flow field of different LW scenarios to improve our understanding of LW dynamics in streams outside of flume experiments. Finally, although the primary fish species of interest in Mill Creek is Coho Salmon, our results are relevant to other salmonids and non-salmonids that also benefit from reduced
velocity and increased channel bed stability.

**Data availability.** Nays2D predicted distributions of velocity and shear stress are available at the ScholarsArchive@OSU (https://ir.library.oregonstate.edu/concern/datasets/br86b895f)

**Acknowledgements**

We are grateful to the Fish and Wildlife Habitat in Managed Forests Research Program, the Oregon Watershed Enhancement Board (OWEB), and the Spirit Mountain Community Fund for providing financial support for this research. The authors thank Weyerhaeuser for providing logistical support. We would also like to express gratitude to Jeff Light, Scott Katz, Rich McDonald, Sharon Baywter-Reyes, Desiree Tullos, John Pitlick, and Jason Dunham for many valuable discussions.

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
