# Peer review of "Ouantifying Restoration Success of Wood Introductions to Increase Coho Salmon Winter Habitat"

_Earth Surface Dynamics, 2019_

## Referee Comment (RC1) · Anonymous Referee #1 · 7 May 2019

The manuscript describes an evaluation of a stream restoration project by large wood introduction in three gravel bed streams in the US. A 2D hydrodynamic model is applied, which had been calibrated with field observations. The calibrated model is then applied to study the habitat suitability for a juvenile salmonid species at bankfull discharge. Large wood increases the size of suitable habitat in all three field sites.

I read this manuscript with a lot of interest. I think the subject is very relevant because large wood introduction is a cost-effective stream restoration method, with a lot of benefits for stream ecology. In general, the manuscript is well written and the figures are well prepared. The Introduction contains most relevant information, the methods are clearly described and the results are well presented, as well.

The main critic I have is that the authors only focus on a single discharge (i.e. bankfull)

[Figure]

when presenting the results, while it might not be too difficult to extend the results with other relevant discharge classes as well. When I was reading the Introduction, I had the feeling the authors would go in that direction. On page 2 (lines 22-24) the authors argue that there is a lack of understanding of the effect of large wood on flow conditions under a range of discharges. So why are only results shown for bankfull discharge conditions and not for other conditions? As far as I understand it well, the model was calibrated for several discharge levels (Table 2). So the model calibration would not put limitations for model application at other than bankfull discharge conditions. Furthermore, in this age of abundant computational resources, I would never argue that additional model runs are not possible because of computational costs. Hence, I suggest to extend the results with other relevant discharge conditions to increase the implications of large wood introduction on habitat suitability for the Coho Salmon.

Overall, I think this manuscript has potential to be a valuable addition to the literature, but some works is still required to make it acceptable for publication. Below I have provided general and specific comments to the text.

**General comments**

1. The Introduction is mainly focused on the effect of large wood on streams in the Pacific Northwest (US). In Europe (and most likely also in other continents) wood is also used in stream restoration, which deserves some attention as well. I suggest to at least add some references to studies where wood is used, not only to improve the habitat conditions for fish, but also to improve conditions for macroinvertebrates.

2. Throughout the manuscript the authors use v and $\tau$ to refer to velocity and stress. Sometimes this results in sentences like "...depth-averaged flow v and shear $\tau$..." (Page 12, line 9), which may be difficult to read for readers without much knowledge in hydraulics. Therefore, I suggest to write "velocity" and "stress" in full where possible.

**Specific comments**

- Page 2, line 34: From "Our objective...". I suggest to start a new paragraph here and first summarize in 1-2 sentences the main limitations of previous research, followed by the objective.

- Page 3, line 12: It is more common to characterize annual precipitation sum in mm, than in cm.

- Page 4, lines 9-11: How was the discharge for the depth-discharge rating curves determined? Through measurements or modelling? Please clarify in the text.

- Page 6, line 32: How were these flow velocity measurements performed? This is not mentioned in the text, please clarify.

- Page 8, lines 5-6: The authors mean that the velocity distribution was more homogeneous before LW introduction and more heterogeneous after LW introduction? Please clarify in the text.

- Page 9, lines 4-6: I suggest to show the percentage increase or decrease, which is more consistent with the previous sentences.

- Page 10, lines 13-14: The authors refer to Fig. 6, but the spatial changes are shown in Fig. 5.

- Page 10, lines 14-17: These sentences are somewhat confusing. The authors are referring to a number of observations, but do you mean simulation results? Also, the results do depend on the chosen transport threshold, hence, the word

"independent" should be "dependent", right? I also would not use the term "significant" in this context, since most readers associate it with statistical significance. In general, the authors are discussing the results here, maybe better to move this to the Discussion section.

- Page 11, lines 8-11: The fitted gamma parameter values are not shown. I suggest to add these values to each of the panels of Fig. 6.

- Page 12, line 7: Please add "in" between "increases" and "the heterogeneity".

- Page 14, lines 2-5: The authors refer here to "small reaches", do you mean "narrow"? Please clarify in the text.

**Figures and Tables**

- Figure 1: I suggest to use some colors to indicate the wood and WSE rulers. Or maybe use a solid black line for the wood, instead of the pattern fill.

---

## Referee Comment (RC2) · Anonymous Referee #2 · 29 May 2019

General comments

The manuscript details a study built around stream restoration efforts, and aims to evaluate the effects that large wood placement has on hydraulic habitat for fish. To this end, the authors apply a 2-dimensional hydraulic model, calibrated based on field observations. To assess the relevance of altered channel hydraulics for fish habitat, modeled flow characteristics are linked with empirical information on fish swimming performance and bed material size (to assess its mobility).

I agree with the authors that better understanding of the hydraulic effects that large wood has on stream processes is an important subject. From the basic science point of view, this topic is of interest because large wood is a key driver of many physical and biological processes in river ecosystems. Likewise, this topic is also critical from the

applied river science perspective, because large wood placement to enhance fish habitat is, by far, the most common channel restoration activity (at least in the geographical regions I am most familiar with).

The methodology applied in this study seems to be generally robust, although some additional information on model limitations and uncertainties would be desirable, to provide readers with more complete information. Similarly, interpretations and conclusions appear to be supported by the data, but I would encourage the authors to elaborate further on this in the context of model limitations. From the technical point of view, the manuscript is written well and has good quality figures that convey key results effectively. However, the manuscript would benefit from exploring in more depth some "pockets" of relevant literature to better contextualize the results. Below, I expand on all the above concerns in more length and give some suggestions.

Recommendations:

1. Methodology and interpretations. Numerical modeling of flow around large wood is a highly challenging task and there have been relatively few attempts to resolve such flow field in 3D. Thus, in my view, the 2D approach adopted by the authors can be still considered current research standard (e.g., Hafs et al., 2014; Wall et al., 2016). However, as the authors acknowledge, there are clear issues related to modeling highly complex, 3D flow using depth-averaged model and substantial errors can be expected as some assumptions are violated, at least locally (e.g., Shen and Diplas, 2008). Given the importance of this issue, I think the authors devote too little discussion to this limitation.

I would recommend that the authors discuss how the modeled flow field resembles or deviates from the patterns observed in various field studies (Daniels and Rhoads, 2003; 2004a; 2004b; Manners et al., 2007) or in experimental setting (see references below). What are the key uncertainties in the predicted flow given what we know about 3D flow structure around such obstructions? What are the implications for the predicted

hydraulic habitat? After all, fish utilize 3D habitat and can adjust their vertical position in the water column. While these uncertainties certainly do not constitute a disqualifying problem, in my opinion, they need to be signaled to readers more clearly and in more detail, so that they can more readily formulate their own judgement regarding the results.

It would be also informative to know how much of the changes in flow hydraulics (and habitat) occur in close proximity to large wood, where errors are likely large, and how much in the far field, away from the wood? For example, is there a way to plot errors in velocity (modeled-observed) against distance from wood, to get a sense of the spatial extent of the zone where flow properties are not captured well? For example, (Xu and Liu, 2017) showed that flow field predictions away from large wood may be reasonable even if a simple solid body representation is chosen.

2. Literature. The authors generally did a good job presenting most relevant literature but I feel that it is slightly less comprehensive on the numerical modeling side. Because modeling is at the core of this paper, I think the paper would benefit from exploring this literature both for providing background to the reader and for contextualizing the results. Allen and Smith (2012) and (Xu and Liu 2016; 2017) are examples of good recent references to cutting-edge approaches to tackle the challenge of modeling flow near complex features like large wood.

In addition, hydraulics of large wood, and particularly engineered log jams (which tend to have simple geometry) bear some similarities to flow around abutments and spur dikes. These parallels have been widely recognized and utilized in the geomorphic literature, e.g. see Abbe and Montgomery (1996) or Buffington et al. (2002). This kind of flow obstructions has been extensively modeled using CFD and engineering literature can serve as a rich source to draw upon in research on large wood; such modeling efforts have also been carried out by river scientists studying restoration structures such as deflectors – see work of Biron and colleagues (Biron et al. 2009; 2012).
3. Uncertainties. The authors should be commended for evaluating model performance on a number of occasions. I think it may be useful to provide more information about this important step of CFD model application. For example, I suggest that the authors consider providing information on the number of measurements used for evaluation (e.g., number of velocity measurements) and the slopes of the regression lines. The latter might be relevant, because bias in modeled velocity, relative to observed velocity, can lead to over- or underestimated heterogeneity in the modeled flow field. For example, if the slope in a modeled conditions prior to wood placement is 0.6 and, after large wood placement is 0.8, this needs to be taken into account when comparing the differences in complexity of flow field due to large wood placement.

I even wonder if it may make sense to carry out a separate comparison based on field data alone and then another one based on the modeled data, and see if those two results converge; of course, this is only if there are enough data to run reasonable regressions based on field data alone. I also noted that the reported velocity errors seem much higher than those for WSE, which should also be highlighted in the discussion, since velocity affects both aspects of habitat that are of interest here (bed shear stress is a function of velocity squared). The errors are within the range reported in the literature, so the magnitude of errors itself is not alarming, but this issue should be communicated clearly in the text. Also, personally I find that showing the data graphically is often as informative as reporting statistics (or more). I leave it to the authors to decide on the most appropriate course of action.

4. Flow event choice. Lastly, I would recommend that the authors further clarify the ecological relevance of bankfull flow for answering their research question.Why was it chosen for this paper out of a wide range of discharges a rainfall-dominated stream may experience during the winter season? Of course, this does seem like an intuitive choice for bed mobility modeling. However, it is slightly less clear why that would be the key flow for fish. Bankfull flow in wet coastal streams in Oregon has, on average, recurrence interval of ∼1.2 years (Castro and Jackson 2001) and in pluvial hydrological
regime flows in excess of that discharge probably last a few days per year. If bankfull flow is critical because of limited flow refugia, or was chosen because of its relevance for sediment transport and because changes in habitat patterns at lower discharges are similar, that should be clearly conveyed in the manuscript. I think it is important for readers to be able to understand broader importance of the reported results, how they extend beyond just a single flow event.

Minor comments & suggestions:

p. 1, line 25-27: LW also influences bed texture – consider citing work of (Buffington and Montgomery 1999).

p. 2, line 5-7: interesting work on LW removal effects by R.D. Smith and colleagues (Smith et al. 1993a, 1993b)

p. 2, line 8-12: I think the clarity of this paragraph would be improved if the authors added a sentence that stated clearly that low velocity habitat is critical for overwinter juvenile survival. This is perhaps a minor point but for the readership of ESD not familiar with fish ecology can be helpful in following the logical flow of this argument (overwinter survival of juveniles key for population viability & low velocity important for juvenile survival => low velocity habitat critical for population recovery).

p. 2, line 16: work of Sommer et al. and Jeffres et al., while undoubtedly interesting and relevant, should be cited with caution in this context, since it was conducted on a very different river system in different climate (floodplains of larger rivers in California Central Valley).

p. 2, line 31-34: I want to point out excellent work by A. Finstad and colleagues on the importance of bed shelters for salmonids, although, of course, there may be some differences betwee Atlantic salmon and Coho (Finstad et al. 2007; 2009)

p. 3, line 12: suggest reporting in units of mm or m, not cm.

p. 3, line 14: how, specifically, are the study reaches geomorphically distinct? Please

clarify

Table 1: what is bankfull area?

p. 6, line 20: the equation (3) defines Cf parameter, then text (e.g., line 27 on that page) refers to Cd – are those the same? Or is this just a typo? Please fix or clarify.

p. 7, line 1-3: given that large wood is at the heart of this study, it actually would be interesting to evaluate also flow field around wood. Once again, poor performance in those areas is to be expected and, in my view, does not disqualify this or any other similar work using 2D, given very limited alternatives, but it would be informative to know the magnitude of errors an spatial extent of the zone within which flow parameters are not modeled reliably. For example, one could evaluate model prediction near and away from LW, or compare evaluations including and excluding near-LW data points.

p.7, line 7: depth threshold of 0.1m seems somewhat high for juvenile Coho (they can certainly swim in shallower flows). But perhaps there is also another reason/criterion why this cutoff was chosen?

p. 9, line 12: perhaps "robustness" not "resiliency"?

p. 11, line 9: could the authors clarify whether/how gamma distribution was fitted in cases of bimodal data?

p. 12, line 4 (and elsewhere): I would encourage the authors to refer to "modeled" or "simulated" habitat rather than habitat. This may seem like hairsplitting but I think it would be prudent to emphasize that these are model predictions rather than empirical data.

p. 15, line 8: "processed" not "process"

Throughout the paper, the authors use v and u for downstream and cross-stream components of the velocity vector (e.g. equation (1) but later v also comes up to describe swimming velocity criteria for fish. The authors should be careful here to avoid using

the same symbol for different variables – please fix or clarify.

In sum, I want to emphasize once again that I believe that, upon revisions, this manuscript could be a valuable contribution to the literature. It focuses on important subject within the field of ecgeomorphology and the methodological approach it adopts, despite some limitations, is scientifically defensible and in line with current research practice. As a result, I believe the reported results are robust and will be of interest to the readership, especially researchers interested in topics at the intersection of earth surface processes and ecology. I look forward to seeing authors' responses as well as the revised manuscript.

References

Abbe, T.B., and Montgomery, D.R. 1996. Large woody debris jams, channel hydraulics and habitat formation in large rivers. Regulated Rivers: Research & Management 12(2‐3): 201–221.

Allen, J.B., and Smith, D.L. 2012. Characterizing the impact of geometric simplification on large woody debris using CFD. International Journal of Hydraulic Engineering 1(2): 1–14.

Biron, P.M., Carré, D.M., Gaskin, S.J., and Brebbia, C. 2009. Hydraulics of stream deflectors used in fish-habitat restoration schemes. WIT Transactions on Ecology and the Environment 124: 305–314.

Biron, P.M., Carver, R.B., and Carré, D.M. 2012. Sediment transport and flow dynamics around a restored pool in a fish habitat rehabilitation project: Field and 3D numerical modelling experiments. River Research and Applications 28(7): 926–939.

Buffington, J.M., Lisle, T.E., Woodsmith, R.D., and Hilton, S. 2002. Controls on the size and occurrence of pools in coarse-grained forest rivers. River Res. Applic. 18(6): 507–531. doi:10.1002/rra.693.

Buffington, J.M., and Montgomery, D.R. 1999. Effects of hydraulic roughness on

surface textures of gravel-bed rivers. Water Resour. Res. 35(11): 3507–3521. doi:10.1029/1999WR900138.

Castro, J.M., and Jackson, P.L. 2001. Bankfull discharge recurrence intervals ani) regional hydraulic geometry relationships: patterns in the Pacific Northwest, USA. JAWRA Journal of the American Water Resources Association 37(5): 1249–1262.

Daniels, M.D., and Rhoads, B.L. 2003. Influence of a large woody debris obstruction on three-dimensional flow structure in a meander bend. Geomorphology 51(1–3): 159–173.

Daniels, M.D., and Rhoads, B.L. 2004a. Effect of large woody debris configuration on three‐dimensional flow structure in two low‐energy meander bends at varying stages. Water Resources Research 40(11).

Daniels, M.D., and Rhoads, B.L. 2004b. Spatial pattern of turbulence kinetic energy and shear stress in a meander bend with large woody debris. Riparian vegetation and fluvial geomorphology: 87–98.

Finstad, A.G., Einum, S., Forseth, T., and Ugedal, O. 2007. Shelter availability affects behaviour, size-dependent and mean growth of juvenile Atlantic salmon. Freshwater Biology 52(9): 1710–1718. doi:10.1111/j.1365-2427.2007.01799.x.

Finstad, A.G., Einum, S., Ugedal, O., and Forseth, T. 2009. Spatial distribution of limited resources and local density regulation in juvenile Atlantic salmon. Journal of Animal Ecology 78(1): 226–235.

Hafs, A.W., Harrison, L.R., Utz, R.M., and Dunne, T. 2014. Quantifying the role of woody debris in providing bioenergetically favorable habitat for juvenile salmon. Ecological modelling 285: 30–38.

Manners, R.B., Doyle, M.W., and Small, M.J. 2007. Structure and hydraulics of natural woody debris jams. Water Resources Research 43(6). doi:10.1029/2006WR004910.

**ESurfD**
[Figure]

Shen, Y., and Diplas, P. 2008. Application of two- and three-dimensional computational fluid dynamics models to complex ecological stream flows. Journal of Hydrology 348(1–2): 195–214. doi:10.1016/j.jhydrol.2007.09.060.

Smith, R.D., Sidle, R.C., and Porter, P.E. 1993a. Effects on bedload transport of experimental removal of woody debris from a forest gravel‐bed stream. Earth Surface Processes and Landforms 18(5): 455–468.

Smith, R.D., Sidle, R.C., Porter, P.E., and Noel, J.R. 1993b. Effects of experimental removal of woody debris on the channel morphology of a forest, gravel-bed stream. Journal of Hydrology 152(1): 153–178. doi:10.1016/0022-1694(93)90144-X.

Wall, C.E., Bouwes, N., Wheaton, J.M., Bennett, S.N., Saunders, W.C., McHugh, P.A., and Jordan, C.E. 2016. Design and monitoring of woody structures and their benefits to juvenile steelhead (Oncorhynchus mykiss) using a net rate of energy intake model. Canadian Journal of Fisheries and Aquatic Sciences 74(5): 727–738.

Xu, Y., and Liu, X. 2016. 3D computational modeling of stream flow resistance due to large woody debris. In Proc. River Flow. pp. 2346–2353.

Xu, Y., and Liu, X. 2017. Effects of Different In-Stream Structure Representations in Computational Fluid Dynamics Models—Taking Engineered Log Jams (ELJ) as an Example. Water 9(2): 110.

---

## Author Comment (AC1) · 1 Jul 2019

**Reply to Anonymous Referee #1**

The manuscript describes an evaluation of a stream restoration project by large wood introduction in three gravel bed streams in the US. A 2D hydrodynamic model is applied, which had been calibrated with field observations. The calibrated model is then applied to study the habitat suitability for a juvenile salmonid species at bankfull discharge. Large wood increases the size of suitable habitat in all three field sites.

I read this manuscript with a lot of interest. I think the subject is very relevant because large wood introduction is a cost-effective stream restoration method, with a lot of benefits for stream ecology. In general, the manuscript is well written and the figures are well prepared. The Introduction contains most relevant information, the methods are clearly described and the results are well presented, as well.

The main critic I have is that the authors only focus on a single discharge (i.e. bankfull) when presenting the results, while it might not be too difficult to extend the results with other relevant discharge classes as well. When I was reading the Introduction, I had the feeling the authors would go in that direction. On page 2 (lines 22-24) the authors argue that there is a lack of understanding of the effect of large wood on flow conditions under a range of discharges. So why are only results shown for bankfull discharge conditions and not for other conditions? As far as I understand it well, the model was calibrated for several discharge levels (Table 2). So the model calibration would not put limitations for model application at other than bankfull discharge conditions. Furthermore, in this age of abundant computational resources, I would never argue that additional model runs are not possible because of computational costs. Hence, I suggest to extend the results with other relevant discharge conditions to increase the implications of large wood introduction on habitat suitability for the Coho Salmon.

**Reply**: We appreciate this comment, which was also raised by the other reviewer. Given the Nays2D is unsteady we actually run the model for 35–45 hour long hydrographs that peaked around bankfull but included a wide range of flows in all sites.  We made this clearer in the methods (P5, L17–20; P7, 28–30). Based on these simulations we now include a section in the results highlighting changes in simulated habitat availability before and after the addition of LW during the whole hydrograph duration (see section 3.3., Figure 7 and Table 3).

Overall, I think this manuscript has potential to be a valuable addition to the literature, but some works is still required to make it acceptable for publication. Below I have provided general and specific comments to the text.

**Reply**:  We really appreciate your careful review.

**General comments**

The Introduction is mainly focused on the effect of large wood on streams in the Pacific Northwest (US). In Europe (and most likely also in other continents) wood is also used in stream restoration, which deserves some attention as well. I suggest to at least add some references to studies where wood is used, not only to improve the habitat conditions for fish, but also to improve conditions for macroinvertebrates.

**Reply**:  We appreciate the suggestion. We added a paper about wood in  European rivers (Kail, 2003) and two papers about the importance of wood for macroinvertebrates (Gerhard & Reich, 2000; Jahnig & Lorenz, 2008) (P1, L24; P1, L30; P2, L1).

Throughout the manuscript the authors use v and τ to refer to velocity and stress. Sometimes this results in sentences like "...depth-averaged flow v and shear τ ..." (Page 12, line 9), which may be difficult to read for readers without much knowledge in hydraulics. Therefore, I suggest to write "velocity" and "stress" in full where possible.

**Reply**: We agree. We eliminated most of the "v" and "t" to improve readability throughout the text.

**Specific comments**

• Page 2, line 34: From "Our objective...". I suggest to start a new paragraph here and first summarize in 1-2 sentences the main limitations of previous research, followed by the objective.

**Reply**: As suggested, we added a new paragraph clearly stating the limitation of previous efforts before stating our objective (P3, L 11–16).

• Page 3, line 12: It is more common to characterize annual precipitation sum in mm, than in cm.

**Reply**: Done (P3, L24).

• Page 4, lines 9-11: How was the discharge for the depth-discharge rating curves determined? Through measurements or modelling? Please clarify in the text.

**Reply**: We added information about how we developed the stage discharge relations: Discharge was measured using the velocity-area method (Dingman, 2002) using a Hack FH950 Portable Velocity meter and depth-discharge rating curves were developed based on 9-10 discharge measurements per site (P4, L13–14).

• Page 6, line 32: How were these flow velocity measurements performed? This is not mentioned in the text, please clarify.

**Reply**: We have clarified in the text that these 13–24 velocity measurements per site were taken across the stream (Figure 2, Table 2) for 2-3 flow levels (P7, L22).

• Page 8, lines 5-6: The authors mean that the velocity distribution was more homogeneous before LW introduction and more heterogeneous after LW introduction? Please clarify in the text.

**Reply**: We have added some text clarifying that the velocity distributions were more homogenous before the LW additions (P8, L26).

• Page 9, lines 4-6: I suggest to show the percentage increase or decrease, which is more consistent with the previous sentences.

**Reply**: The suggested changed was implemented (P9, L15–16).

• Page 10, lines 13-14: The authors refer to Fig. 6, but the spatial changes are shown in Fig. 5.

**Reply**: Yes, you are correct, thank you (P11, L8)

• Page 10, lines 14-17: These sentences are somewhat confusing. The authors are referring to a number of observations, but do you mean simulation results? Also, the results do depend on the chosen transport threshold, hence, the word "independent" should be "dependent", right? I also would not use the term "significant" in this context, since most readers associate it with statistical significance. In general, the authors are discussing the results here, maybe better to move this to the Discussion section.

**Reply**: We agree that these sentences do not belong in the results section.  We decided to eliminate them as they do not add much to our findings.

• Page 11, lines 8-11: The fitted gamma parameter values are not shown. I suggest to add these values to each of the panels of Fig. 6.

**Reply**: We appreciated the suggestion. The values have been added to the figure.

• Page 12, line 7: Please add "in" between "increases" and "the heterogeneity".

**Reply**: Done (P14, 4).

• Page 14, lines 2-5: The authors refer here to "small reaches", do you mean "narrow"? Please clarify in the text.

**Reply**: We mean small not only in the sense of narrow but also smaller in terms of having less drainage area and thus less discharge. We have clarified this in the text (P16, L21 –22).

**Figures and Tables**

• Figure 1: I suggest to use some colors to indicate the wood and WSE rulers. Or maybe use a solid black line for the wood, instead of the pattern fill.

**Reply**: We believe you are referring to figure 2 here.  We changed the color of the Wood pieces as suggested.

**References mentioned in the reply.**

Dingman, S. L. (2002). *Physical hydrology*. Upper Saddle River, N.J.: Prentice Hall.
Gerhard, M., & Reich, M. (2000). Restoration of streams with large wood: Effects of accumulated and built-in wood on channel morphology, habitat diversity and aquatic fauna. *International Review of Hydrobiology, 85*(1), 123-137. doi:10.1002/(sici)1522-2632(200003)85:1<123::aid-iroh123>3.3.co;2-k
Jahnig, S. C., & Lorenz, A. W. (2008). Substrate-specific macroinvertebrate diversity patterns following stream restoration. *Aquatic Sciences, 70*(3), 292-303. doi:10.1007/s00027-008-8042-0
Kail, J. (2003). Influence of large woody debris on the morphology of six central European streams. *Geomorphology, 51*(1), 207-223. doi:https://doi.org/10.1016/S0169-555X(02)00337-9

---

## Author Comment (AC2) · 1 Jul 2019

**Reply to Anonymous Referee #2**

General comments

The manuscript details a study built around stream restoration efforts, and aims to evaluate the effects that large wood placement has on hydraulic habitat for fish. To this end, the authors apply a 2-dimensional hydraulic model, calibrated based on field observations. To assess the relevance of altered channel hydraulics for fish habitat, modeled flow characteristics are linked with empirical information on fish swimming performance and bed material size (to assess its mobility).

I agree with the authors that better understanding of the hydraulic effects that large wood has on stream processes is an important subject. From the basic science point of view, this topic is of interest because large wood is a key driver of many physical and biological processes in river ecosystems. Likewise, this topic is also critical from the applied river science perspective, because large wood placement to enhance fish habitat is, by far, the most common channel restoration activity (at least in the geographical regions I am most familiar with).

The methodology applied in this study seems to be generally robust, although some additional information on model limitations and uncertainties would be desirable, to provide readers with more complete information. Similarly, interpretations and conclusions appear to be supported by the data, but I would encourage the authors to elaborate further on this in the context of model limitations. From the technical point of view, the manuscript is written well and has good quality figures that convey key results effectively. However, the manuscript would benefit from exploring in more depth some "pockets" of relevant literature to better contextualize the results. Below, I expand on all the above concerns in more length and give some suggestions.

**Recommendations:**

Methodology and interpretations. Numerical modeling of flow around large wood is a highly challenging task and there have been relatively few attempts to resolve such flow field in 3D. Thus, in my view, the 2D approach adopted by the authors can be still considered current research standard (e.g., Hafs et al., 2014; Wall et al., 2016). However, as the authors acknowledge, there are clear issues related to modeling highly complex, 3D flow using depth-averaged model and substantial errors can be expected as some assumptions are violated, at least locally (e.g., Shen and Diplas, 2008). Given the importance of this issue, I think the authors devote too little discussion to this limitation.

I would recommend that the authors discuss how the modeled flow field resembles or deviates from the patterns observed in various field studies (Daniels and Rhoads, 2003; 2004a; 2004b; Manners et al., 2007) or in experimental setting (see references below). What are the key uncertainties in the predicted flow given what we know about 3D flow structure around such obstructions? What are the implications for the predicted hydraulic habitat? After all, fish utilize 3D habitat and can adjust their vertical position in the water column. While these uncertainties certainly do not constitute a disqualifying problem, in my opinion, they need to be signaled to readers more clearly and in more detail, so that they can more readily formulate their own judgement regarding the results.

It would be also informative to know how much of the changes in flow hydraulics (and habitat) occur in close proximity to large wood, where errors are likely large, and how much in the far field, away from the wood? For example, is there a way to plot errors in velocity (modeled-observed) against distance from wood, to get a sense of the spatial extent of the zone where flow properties are not captured well?

For example, (Xu and Liu, 2017) showed that flow field predictions away from large wood may be reasonable even if a simple solid body representation is chosen.

**Reply**: We appreciate your suggestion and agree. We now acknowledge some of the potential issues associated with 2D modelling of flow around obstacles in the discussion section (P 15, L1-16). We also added that the flow around the LW jams will change over time given channel adjustment and the addition of smaller pieces of wood to the jams over time (P15, 22–26). We indicate how our predictions seem to resemble 3D predictions while acknowledging that we lack information to assess the performance of the model near the LW obstacles. We agree that assessing the model predictions based on multiple velocity measurements taken at different distances from the LW would be very informative, but we lack such data. A detailed assessment of this kind is challenging at the reach scale during winter flows because the reaches are not wadable. The velocity measurements we have were collected at a cross-section per reach (Figure 2) 7–20 meters away from the LW additions. We have added to figure 2 the cross-sections in which we collected the velocity measurements.

Literature. The authors generally did a good job presenting most relevant literature but I feel that it is slightly less comprehensive on the numerical modeling side. Because modeling is at the core of this paper, I think the paper would benefit from exploring this literature both for providing background to the reader and for contextualizing the results. Allen and Smith (2012) and (Xu and Liu 2016; 2017) are examples of good recent references to cutting-edge approaches to tackle the challenge of modeling flow near complex features like large wood. In addition, hydraulics of large wood, and particularly engineered log jams (which tend to have simple geometry) bear some similarities to flow around abutments and spur dikes. These parallels have been widely recognized and utilized in the geomorphic literature, e.g. see Abbe and Montgomery (1996) or Buffington et al. (2002). This kind of flow obstructions has been extensively modeled using CFD and engineering literature can serve as a rich source to draw upon in research on large wood; such modeling efforts have also been carried out by river scientists studying restoration structures such as deflectors – see work of Biron and colleagues (Biron et al. 2009; 2012).

**Reply**: We appreciate the suggestion a paragraph was added to the introduction (P2, L27–33) providing background about the use of CFD models to simulate flow filed conditions around wood.

Uncertainties. The authors should be commended for evaluating model performance on a number of occasions. I think it may be useful to provide more information about this important step of CFD model application. For example, I suggest that the authors consider providing information on the number of measurements used for evaluation (e.g., number of velocity measurements) and the slopes of the regression lines. The latter might be relevant, because bias in modeled velocity, relative to observed velocity, can lead to over- or underestimated heterogeneity in the modeled flow field. For example, if the slope in a modeled conditions prior to wood placement is 0.6 and, after large wood placement is 0.8, this needs to be taken into account when comparing the differences in complexity of flow field due to large wood placement.
I even wonder if it may make sense to carry out a separate comparison based on field data alone and then another one based on the modeled data, and see if those two results converge; of course, this is only if there are enough data to run reasonable regressions based on field data alone. I also noted that the reported velocity errors seem much higher than those for WSE, which should also be highlighted in the discussion, since velocity affects both aspects of habitat that are of interest here (bed shear stress is a function of velocity squared). The errors are within the range reported in the literature, so the magnitude of errors itself is not alarming, but this issue should be communicated clearly in the text.

Also, personally I find that showing the data graphically is often as informative as reporting statistics (or more). I leave it to the authors to decide on the most appropriate course of action.

**Reply**: The number of observations of water surface elevation and velocity used to evaluate model performance and the slope of the water surface before and after the wood placement were added to Table 2.  The calculated WSE slopes are within 10% of the observed slopes indicating strong performance of the model.  The velocity observations pre and post wood were used as an additional check. However, given how difficult (dangerous) it is to collect velocity measurements at high flows our calibration relied strongly in the WSE observations. We clarified this in the methods (P7, L20–22) and in the discussion (P15, L12–16).  We found that that the mean WSE slopes are higher post wood than pre wood.  This change in slope is a reflection of the effects of the wood in the flow field (P9, L1; P11, L10–11; P15, 32–34).

Flow event choice. Lastly, I would recommend that the authors further clarify the ecological relevance of bankfull flow for answering their research question. Why was it chosen for this paper out of a wide range of discharges a rainfall-dominated stream may experience during the winter season? Of course, this does seem like an intuitive choice for bed mobility modeling. However, it is slightly less clear why that would be the key flow for fish. Bankfull flow in wet coastal streams in Oregon has, on average, recurrence interval of ~1.2 years (Castro and Jackson 2001) and in pluvial hydrological regime flows in excess of that discharge probably last a few days per year. If bankfull flow is critical because of limited flow refugia, or was chosen because of its relevance for sediment transport and because changes in habitat patterns at lower discharges are similar, that should be clearly conveyed in the manuscript. I think it is important for readers to be able to understand broader importance of the reported results, how they extend beyond just a single flow event.

**Reply**: We appreciate this comment, which was also raised by the other reviewer. Given the Nays2D is unsteady we actually run the model for 35–45 hour long hydrographs that peaked around bankfull but included a wide range of flows in all sites.  We made this clearer in the methods (P7, L28–30). Based on these simulations we now include a section in the results highlighting changes in simulated habitat availability before and after the addition of LW during the whole hydrograph duration (see section 3.3., Figure 7 and Table 3).

**Minor comments & suggestions:**

p. 1, line 25-27: LW also influences bed texture – consider citing work of (Buffington and Montgomery 1999).

**Reply**: We added this reference to the introduction (P1, L25).

p. 2, line 5-7: interesting work on LW removal effects by R.D. Smith and colleagues (Smith et al. 1993a, 1993b)

**Reply**:  We agree, we have added these references (P2, L8).

p. 2, line 8-12: I think the clarity of this paragraph would be improved if the authors added a sentence that stated clearly that low velocity habitat is critical for overwinter juvenile survival. This is perhaps a minor point but for the readership of ESD not familiar with fish ecology can be helpful in following the

logical flow of this argument (overwinter survival of juveniles key for population viability & low velocity important for juvenile survival => low velocity habitat critical for population recovery).

**Reply:** We appreciate this suggestion and the paragraph has been edited throughout to improve clarity and logical flow (P2, L9–16).

p. 2, line 16: work of Sommer et al. and Jeffres et al., while undoubtedly interesting and relevant, should be cited with caution in this context, since it was conducted on a very different river system in different climate (floodplains of larger rivers in California Central Valley).

**Reply**: We agree with you. We decided to remove these two references.

p. 2, line 31-34: I want to point out excellent work by A. Finstad and colleagues on the importance of bed shelters for salmonids, although, of course, there may be some differences between Atlantic salmon and Coho (Finstad et al. 2007; 2009)

**Reply:** We appreciate this suggestion, but the work by Finstad and colleagues on Atlantic Salmon does not directly translate to Coho Salmon, which are not as strongly associated with the substrate during normal winter flows, therefore we did not add these references.

p. 3, line 12: suggest reporting in units of mm or m, not cm.

**Reply**: Done, (P3, L24).

p. 3, line 14: how, specifically, are the study reaches geomorphically distinct? Please clarify Table 1: what is bankfull area?

**Reply**: We change this sentence eliminating the notion that the reaches are located in distinct geomorphology. The most relevant point here is they are all low gradient and fish bearing (P3, L25). Bank full area is the cross-sectional area at bankfull level. We added the word cross-sectional (Table 1) in an effort to make this clearer.

p. 6, line 20: the equation (3) defines Cf parameter, then text (e.g., line 27 on that page) refers to Cd – are those the same? Or is this just a typo? Please fix or clarify.

**Reply**: Yes that was a typo thank you (P7, L15)

p. 7, line 1-3: given that large wood is at the heart of this study, it actually would be interesting to evaluate also flow field around wood. Once again, poor performance in those areas is to be expected and, in my view, does not disqualify this or any other similar work using 2D, given very limited alternatives, but it would be informative to know the magnitude of errors an spatial extent of the zone within which flow parameters are not modeled reliably. For example, one could evaluate model prediction near and away from LW, or compare evaluations including and excluding near-LW data points.

**Reply**: Agreed that this would be an interesting line of inquiry, unfortunately, we do not have the velocity measurement data to look into this and through calibration. We clarified this in the methods (P7, L20–21) and in the discussion (P15, L12–16).

p.7, line 7: depth threshold of 0.1m seems somewhat high for juvenile Coho (they can certainly swim in shallower flows). But perhaps there is also another reason/criterion why this cutoff was chosen?

**Reply:** While juvenile Coho can certainly swim in very shallow areas, they are seldom found in water less than 0.1 m deep during the winter (Bustard and Narver 1975a). The text was edited to clarify this point, and the reference above was added to the text (P8, L1).

p. 9, line 12: perhaps "robustness" not "resiliency"?

**Reply**: We agree the changed was made (P10, L1).

p. 11, line 9: could the authors clarify whether/how gamma distribution was fitted in cases of bimodal data?

**Reply**: We follow the methodology describe in (Segura & Pitlick, 2015). The parameters of the gamma function that best fitted the distributions were found by systematically varying the $\alpha$ parameter between 0 and 60 in increments of 0.01 (i.e., total 6000 $\alpha$ values tested) and finding the parameter values that yielded the lowest overall $\chi$2 score. We added the mention reference. Given that we are trying to make predictions based in this fits but rather to illustrate the changes in shape of distributions of shear stress we believe there is no need to provide more details here. However, here is a figure of the fits for your review.

[Figure]

Figure R1: Gamma fits to the distributions of shear stress before (A-C) and after (D-F) the additions of LW in sites 1 (A,D), Site 2(B,E) and site 3 (C,F).

p. 12, line 4 (and elsewhere): I would encourage the authors to refer to "modeled" or "simulated" habitat rather than habitat. This may seem like hairsplitting but I think it would be prudent to emphasize that these are model predictions rather than empirical data.

**Reply**: We agree. We made changes accordingly.

p. 15, line 8: "processed" not "process"

**Reply**: The wording was changed (P17, L22–23)

Throughout the paper, the authors use v and u for downstream and cross-stream components of the velocity vector (e.g. equation (1) but later v also comes up to describe swimming velocity criteria for fish. The authors should be careful here to avoid using the same symbol for different variables – please fix or clarify.

**Reply:** We agree, we changed the notation for the cross-stream velocity component (P6, L26)

In sum, I want to emphasize once again that I believe that, upon revisions, this manuscript could be a valuable contribution to the literature. It focuses on important subject within the field of ecgeomorphology and the methodological approach it adopts, despite some limitations, is scientifically defensible and in line with current research practice. As a result, I believe the reported results are robust and will be of interest to the readership, especially researchers interested in topics at the intersection of earth surface processes and ecology. I look forward to seeing authors' responses as well as the revised manuscript.

**Reply**: We really appreciate your careful and thoughtful review.

**References**

Abbe, T.B., and Montgomery, D.R. 1996. Large woody debris jams, channel hydraulics and habitat formation in large rivers. Regulated Rivers: Research & Management 12(2âAˇR3): 201–221. ˇ

Allen, J.B., and Smith, D.L. 2012. Characterizing the impact of geometric simplification on large woody debris using CFD. International Journal of Hydraulic Engineering 1(2): 1–14.

Biron, P.M., Carré, D.M., Gaskin, S.J., and Brebbia, C. 2009. Hydraulics of stream deflectors used in fish-habitat restoration schemes. WIT Transactions on Ecology and the Environment 124: 305–314.

Biron, P.M., Carver, R.B., and Carré, D.M. 2012. Sediment transport and flow dynamics around a restored pool in a fish habitat rehabilitation project: Field and 3D numerical modelling experiments. River Research and Applications 28(7): 926–939.

Buffington, J.M., Lisle, T.E., Woodsmith, R.D., and Hilton, S. 2002. Controls on the size and occurrence of pools in coarse-grained forest rivers. River Res. Applic. 18(6): 507–531. doi:10.1002/rra.693.

Buffington, J.M., and Montgomery, D.R. 1999. Effects of hydraulic roughness on surface textures of gravel-bed rivers. Water Resour. Res. 35(11): 3507–3521. doi:10.1029/1999WR900138.

Castro, J.M., and Jackson, P.L. 2001. Bankfull discharge recurrence intervals and regional hydraulic geometry relationships: patterns in the Pacific Northwest, USA. JAWRA Journal of the American Water Resources Association 37(5): 1249–1262.

Daniels, M.D., and Rhoads, B.L. 2003. Influence of a large woody debris obstruction on three-dimensional flow structure in a meander bend. Geomorphology 51(1–3): 159– 173.

Daniels, M.D., and Rhoads, B.L. 2004a. Effect of large woody debris configuration on threeâAˇRdimensional flow structure in two lowâ ˇ Aˇ Renergy meander bends at varying ˇ stages. Water Resources Research 40(11).

Daniels, M.D., and Rhoads, B.L. 2004b. Spatial pattern of turbulence kinetic energy and shear stress in a meander bend with large woody debris. Riparian vegetation and fluvial geomorphology: 87–98.

Finstad, A.G., Einum, S., Forseth, T., and Ugedal, O. 2007. Shelter availability affects behaviour, size-dependent and mean growth of juvenile Atlantic salmon. Freshwater Biology 52(9): 1710–1718. doi:10.1111/j.1365-2427.2007.01799.x.

Finstad, A.G., Einum, S., Ugedal, O., and Forseth, T. 2009. Spatial distribution of limited resources and local density regulation in juvenile Atlantic salmon. Journal of Animal Ecology 78(1): 226–235.

Hafs, A.W., Harrison, L.R., Utz, R.M., and Dunne, T. 2014. Quantifying the role of woody debris in providing bioenergetically favorable habitat for juvenile salmon. Ecological modelling 285: 30–38.

Manners, R.B., Doyle, M.W., and Small, M.J. 2007. Structure and hydraulics of natural woody debris jams. Water Resources Research 43(6). doi:10.1029/2006WR004910.

Shen, Y., and Diplas, P. 2008. Application of two- and three-dimensional computational fluid dynamics models to complex ecological stream flows. Journal of Hydrology 348(1–2): 195–214. doi:10.1016/j.jhydrol.2007.09.060.

Smith, R.D., Sidle, R.C., and Porter, P.E. 1993a. Effects on bedload transport of experimental removal of woody debris from a forest gravel⢠Rbed stream. Earth Surface ˘ Processes and Landforms 18(5): 455–468.

Smith, R.D., Sidle, R.C., Porter, P.E., and Noel, J.R. 1993b. Effects of experimental removal of woody debris on the channel morphology of a forest, gravel-bed stream. Journal of Hydrology 152(1): 153–178. doi:10.1016/0022-1694(93)90144-X.

Wall, C.E., Bouwes, N., Wheaton, J.M., Bennett, S.N., Saunders, W.C., McHugh, P.A., and Jordan, C.E. 2016. Design and monitoring of woody structures and their benefits to juvenile steelhead (Oncorhynchus mykiss) using a net rate of energy intake model. Canadian Journal of Fisheries and Aquatic Sciences 74(5): 727–738.

Xu, Y., and Liu, X. 2016. 3D computational modeling of stream flow resistance due to large woody debris. In Proc. River Flow. pp. 2346–2353.

Xu, Y., and Liu, X. 2017. Effects of Different In-Stream Structure Representations in Computational Fluid Dynamics Models Taking Engineered Log Jams (ELJ) as an ˘ Example. Water 9(2): 110.

**References mentioned in the reply.**

Segura, C., & Pitlick, J. (2015). Coupling fluvial-hydraulic models to predict gravel transport in spatially variable flows. *Journal of Geophysical Research-Earth Surface, 120*(5), 834-855. doi:10.1002/2014JF003302